# Whole genome sequencing, characterization and analysis of coronene degrading bacterial strain *Halomonas elongata*

Thasneema Rafic[1], Mohammed Alarawi[2], Omer Salem Alkhnbashi[3,4], Assad Al-Thukair[1], Ajibola H. Okeyode[5], Karthikeyan G.[6], Alexis Nzila[1,7]*

1 Department of Bioengineering, King Fahd University of Petroleum and Minerals, Dhahran, Saudi Arabia, 2 Comparative Genomics and Genetics, King Abdullah University of Science and Technology, King Abdullah University of Science and Technology,Thuwal, Saudi Arabia, 3 Department of Information Computer System, King Fahd University of Petroleum and Minerals, Dhahran, Saudi Arabia, 4 Mohammed Bin Rashid University of Medicine and Health Sciences (MBRU), Dubai Healthcare City, Dubai, United Arab Emirates, 5 College of Petroleum Engineering & Geosciences, King Fahd University of Petroleum and Minerals Dhahran, Saudi Arabia, 6 Department of Microbiology, Kasturba Medical College, Manipal, India, 7 Interdisciplinary Research Center for Membranes and Water Security, King Fahd University of Petroleum and Minerals, Dhahran, Saudi Arabia

* alexisnzila@kfupm.edu.sa

## Abstract

Polycyclic aromatic hydrocarbons (PAHs) are persistent environmental pollutants with significant ecological and health risks. Among them, coronene, a high molecular weight PAH, is particularly resistant to biodegradation due to its complex structure. This study characterizes a halophilic bacterial strain, initially identified as *Halomonas caseinilytica* and later reclassified as *Halomonas elongata*, capable of utilizing coronene as its sole carbon source under high salinity (10% NaCl). Whole genome sequencing using Oxford Nanopore technology (ONT) revealed 4,308 predicted genes, including those linked to hydrocarbon metabolism, stress adaptation, and secondary metabolite biosynthesis. Pathway analysis identified genes associated with xenobiotic degradation, although no canonical coronene specific degradative enzymes were identified, implying that the bacteria may be utilising an alternative or novel pathway. Comparative annotation uncovered operons and enzymes relevant to aromatic compound breakdown. Notably, the presence of ectoine biosynthesis genes suggests a robust osmoadaptation system. Features such as mobile genetic elements and horizontal gene transfer events were also investigated. These findings expand current knowledge on PAH-degrading halophiles and highlight the potential of *H. elongata* in bioremediation of saline and hypersaline environments contaminated with complex hydrocarbons. The study also emphasises the potential of long read sequencing technologies in environmental genomics and bioremediation.

**Data availability statement:** Data is available here: https://www.ebi.ac.uk/ena/browser/view/ERR15384688 and the reference is ERR15384688.

**Funding:** This study was supported by project number INMW2301 by the Interdisciplinary Research Center for Membranes and Water Security (IRC-MWS), King Fahd University of Petroleum and Minerals (KFUPM) Dhahran, Saudi Arabia. There was no additional external funding received for this study.

**Competing interests:** The authors have declared that no competing interests exist.

## Introduction

Polycyclic aromatic hydrocarbons (PAHs) are persistent pollutants that pose serious environmental and health risks due to their toxicity, mutagenicity, and carcinogenicity [1]. These pollutants, commonly originating from the incomplete combustion of organic materials and fossil fuels, are widespread in terrestrial and aquatic ecosystems [2,3].Polycyclic aromatic hydrocarbons (PAHs) can be categorized into two groups. The first group consists of low molecular weight PAHs (LMW-PAHs), which contain two or three aromatic rings. Representative compounds in this group include naphthalene, phenanthrene, and anthracene. The second group comprises high molecular weight PAHs (HMW-PAHs), which contain more than three rings; notable examples include pyrene (four rings), benzo[a]pyrene (five rings), and coronene (seven rings) [3–5]. The literature contains numerous reports on the biodegradation of both LMW-PAHs (e.g., naphthalene, phenanthrene, and anthracene) and HMW-PAHs (e.g., pyrene and benzo[a]pyrene), including studies conducted under thermophilic, halophilic, and anaerobic conditions [1,5–12]. Bacteria capable of degrading PAHs have been identified across a wide range of genera. A study summarizing research on Saudi bacterial strains identified 38 different genera capable of degrading PAHs and other petroleum-derived compounds [13].

Comparatively, limited work has been carried out on the degradation of the HMW-PAHs coronene, due to the complexity of it's structure, which makes it recalcitrant to biodegradation [14]. Three studies have reported the degradation of coronene by strains of *Stenotrophomonas maltophilia* (formerly known as *Burkholderia cepacia*) [15–17]. However, this degradation was observed only in the presence of pyrene, suggesting that these strains may not be capable of utilizing coronene as a sole carbon source. Recently, our research group identified a novel halophilic bacterial strain, *Halomonas caseinilytica* 10SCRN4D, isolated from fuel depots on the campus of King Fahd University of Petroleum and Minerals (Dhahran, Saudi Arabia), which was capable of degrading coronene as the sole carbon source under high salinity conditions (10% NaCl w/v). The discovery of *H. caseinilytica* 10SCRN4D's unique ability to degrade coronene in highly saline environments opens new avenues for research in PAH bioremediation, particularly in marine and hypersaline ecosystems. In addition, this strain was also capable of degrading other high molecular weight PAHs, including benzo[a]pyrene, phenanthrene, and pyrene, indicating a robust and versatile metabolic potential for PAH degradation [18].

To further elucidate the genetic and metabolic mechanisms underlying this exceptional capability, we have conducted a whole genome sequencing analysis of *H. caseinilytica 10SCRN4D*. Whole genome sequencing has proven to be an invaluable tool in understanding the metabolic potential and genetic adaptations of microorganisms involved in biodegradation processes [19]. For instance, the genome analysis of *Mycobacterium sp. strain CH2*, capable of degrading pyrene and benzo[a]pyrene, revealed a complete set of genes responsible for the degradation pathways of these HMW-PAHs [20].

Previous studies relying on Short-read Sequencing Technologies (SRST), such as Illumina, faced challenges in assembling repetitive regions, structural variations, and

long operonic sequences, which are critical for understanding microbial genomic architecture [21]. For example, multiple studies underlined the difficulty in assembling repetitive regions using short reads, leading to fragmented assemblies of bacterial genomes [22–24]. This could be because limited read lengths and lack of paired-end reads pose impediments for assembly software in resolving repeat regions, leading to fragmented assemblies [23]. Another challenge is the inability of short reads to accurately resolve repetitive genomic regions making it arduous to detect genetic variations [21]. In context of our study, where our strain is expected to have a relatively higher GC content as an extremophile, SRST often does not permit to accurately characterize DNA and RNA with extreme GC content, repetitive homologous sequences, or epigenetic modifications, making SRST a poor choice of sequencing technology [24,25]. These shortcomings inevitably restrict functional annotation and hamper the identification of novel pathways. In contrast, Long-read Sequencing Technologies (LRST) has demonstrated superior capabilities. It enables accurate mapping of sequencing reads to reference genomes, facilitates diverse variant detection methodologies, and introduces innovative approaches for characterizing epigenetic diversity [26]. The advancements in sequencing speed and accuracy, alongside the improved quality of bioinformatics analyses, demonstrate the effectiveness of recent technological innovations and their inherent chemical kits [27]. For instance, *Koren et al*. [28] in 2013 demonstrated the power of long reads in resolving complete bacterial genomes, including plasmids and repetitive regions, enhancing our understanding of bacterial evolution and pathogenicity. Another study used LRST for single-cell genomics of uncultivated bacteria, providing insights into microbial dark matter and expanding our knowledge of microbial diversity [29]. In the present study, we use LRST to explore the genetic mechanisms underlying coronene degradation in *H. caseinilytica 10SCRN4D*, we seek to fill the knowledge gap in HMW-PAH biodegradation and offer valuable insights and tools to tackle the enduring issue of PAH contamination across various environmental contexts.

## Materials and methods

### DNA isolation, whole genome sequencing and quality assessment

The strain used in this study, *H. caseinilytica 10SCRN4D* was originally isolated from soil samples collected from a fuel station of King Fahd University of Petroleum and Minerals, as described in *Okeyode et al.(2023)* [18]. In brief, researchers enriched the soil samples under saline conditions using coronene as the only carbon source. This process led to the isolation of this halophilic bacterium, as detailed previously [18].

Bacterial pellet from single colony enrichment was subject to DNA isolation using qiagen MagAttract HMW DNA Kit (Qiagen, Germany). DNA was quantified using Qubit BR Assay Kits (Thermo, USA). 400–500 ng DNA was used to prepare sequencing library for Oxford nano-pore sequencing (ONT) using SQK-LSK109 Ligation Sequencing kit with R9.4.1 flowcell (Oxford Nanopore Technologies, Oxford, UK). The basecalling was performed in realtime using Guppy v5.1.

Bacterial genome assembly and analysis from ONT long reads was performed using the nf-core/bacass pipeline (v2.0.0) using nextflow (v23.04.0) [30]. Raw reads initial quality control and adapter trimming was performed using NanoPlot (v1.38.0) [31] and Porechop (v0.2.4). The *de novo* assembly was utilized Minimap2 (v2.21-r1071) [32] for read alignment and Miniasm (v0.3-r179) [33] and contig generation. The draft assembly was polished using Minimap2, Racon (v1.4.20) [34], and Medaka (v1.4.3) to improve the sequence accuracy. Finally, assembly quality was assessed using QUAST (v5.0.2) [35], and a comprehensive multi-tool report was generated with MultiQC [36].

The completeness of the assembled genome was measured using BUSCO v 5.4.6 (Benchmarking Universal Single-Copy Orthologs) [37], with an E-value cutoff of 0.001 for BLAST searches to ensure high-confidence detection of conserved orthologs while minimizing false positives.

### Strain identification

The thorough analysis of the bacterial genome began strain identification using Kraken2 (v2.1.1), for assigning taxonomic labels and detect contamination [38]. Parameters were set at a 0.5 confidence score to balance sensitivity and specificity.

Additionally, a minimum hit group of 2 was used to avoid weak or ambiguous taxonomic assignments, improving the reliability of strain identification.

## Gene prediction and functional annotation

To ensure comprehensive and accurate gene annotation, three distinct gene prediction tools were employed, each paired with a specific annotation tool. PROKKA [39] was the first tool employed for initial gene prediction, using an e-value cutoff of 1e-06 to ensure highly reliable functional annotations. A 1e-06 cut off was selected to balance specificity and sensitivity as zero is not a valid threshold in BLAST, and this cutoff also minimizes false positives while retaining biological meaningful homologs. The predicted genes were promptly annotated with PROKKA's integrated annotation system. To enhance the depth of functional insights, hypothetical proteins identified by PROKKA were subjected to CDD (Conserved Domains Databases) searches [40,41]. These searches were performed with a stringent e-value threshold of 0.001 and a maximum of 500 hits to allow for the identification of conserved functional domains even in hypothetical proteins, thereby enhancing the depth and biological relevance of the genomic annotations. The functional insights gained from CDD analysis were then integrated with PROKKA's annotations, creating a more comprehensive and detailed overview of the predicted genes' roles and their potential biological significance.

In addition to PROKKA, two other gene prediction tools were employed: PRODIGAL [42] and GeneMarkS2 [43]. The genes predicted by these tools were subsequently annotated using EggNOG-mapper v2, a powerful functional annotation tool. [44]. MAFFT, a multiple sequence alignment program, was utilized to align the gene sequences predicted by all three tools to identify potential discrepancies between the different prediction methods and enhancing the overall accuracy of the annotation process [45]. Finally, gene ontology-based functional annotation was performed using InterProScan and Blast2GO [46].

## Identification of genomic features

To gain insights into gene organization and regulatory mechanisms within the genome, Operon Mapper was utilized to identify potential operons, providing information on gene clustering and regulation [47]. CRISPR (Clustered Regularly Interspaced Short Palindromic Repeats) arrays, known for their role in bacterial immunity and genome editing, were detected using CRISPRCasFinder [48]. This step was crucial to understand the adaptive immune mechanisms of the organism. Additionally, the resistance gene Identifier program of the database CARD (Comprehensive Antibiotic Resistance Database) was used to spot any genes of antibiotic resistance [49].

To further investigate the genome's structure and evolutionary dynamics, RepeatMasker v4.1.5 [50] was employed to identify repeat elements, and RepeatModeler v2.0.5 [51] was used for *de novo* annotation of these repetitive sequences. Additionally, Palindrome v5.0.0.1 [52] was applied to detect inverted repeats, with parameters set as follows: lengths ranging from 10 to 100 base pairs target meaningful structural motifs; a maximum gap of 100 base pairs between repeats to accommodate typical regulatory structures and no mismatches allowed to ensure the identification of exact inverted repeats. Mobile genetic elements, which are pivotal in bacterial evolution and environmental adaptation, were identified using MobileOG-DB with e-value score of 1.0e-05 and k value of 1 that would maximize sensitivity, ensuring the identification of all potentially relevant mobile genetic sequences [53]. Furthermore, potential horizontal gene transfer events, critical for the acquisition of novel traits and rapid adaptation, were detected using Alien Hunter [54].Lastly, the presence of secondary metabolite biosynthesis genes, was identified using the antiSMASH web-based tool [54].

## Pathway analysis

Two complementary approaches were used for pathway analysis: the RAST (Rapid Annotations using Subsystems Technology) server and KAAS (KEGG Automatic Annotation Server). The RAST server was employed to annotate genes based on curated subsystems and protein families [55,56]. KAAS was utilized for functional annotation of genes [57].

KAAS employed the bi-directional best hit (BBH) method, a reliable technique for identifying orthologous relationships. KO (KEGG Orthology) identifiers assigned through this process were subsequently used to automatically generate KEGG pathways and functional classifications.

## Results

### Whole genome sequencing and quality assessment

ONT allowed real time detection and generated long reads of 6 contigs combining to a total length of 3966854 bp, of which the largest contig made up 1702422 bp (maybe repot in Mbp or Kbp). The number of N's per 100 kbp was reported to be zero implying that no ambiguous 'N' bases were in in 100,000 bp (Kbp or Mbp) of the assembly, suggesting an assembly with high sequence continuity without gaps. Table 1 summarises the quality assessment report. Overall, the statistics indicated a high-quality genome assembly with few, large, and contiguous sequences, minimal gaps, and a good representation of the genome's total length (Fig 1). The GC content was reported to be 63.04% which, although within the desirable range of 40%−80%, is still relatively high. Higher GC content often correlates with thermal stability, which suggests that our organism is adapted to high-temperature environments, an information we can confirm from our previous study [18].

The completeness of the genome assembly was then analyzed with BUSCO which validates the quality of genome assemblies based on the presence of highly conserved genes. In Fig 2, our results show that out of a total of 619 BUSCOs searched, 506 were complete. This included 505 that are present as a single copy and 1 that is duplicated. This implies that 81% of orthologs that were found in the genome assembly are intact without missing any important regions. The predominance of single-copy BUSCOs and the minimal duplication suggest that our assembly is accurate and largely free from redundancy or misassembly. Additionally, 72 orthologs were fragmented while 41 were missing from the assembly. Low number of missing genes mean only a small proportion of expected genes are absent. This indicates the genome assembly is mostly comprehensive.

### Taxonomical classification

Taxonomical classification of the sequence was done using Kraken2 software [58] that reclassified the bacteria as *H. elongata* contradicting the previous 16s rRNA based identification of the strain as *H. caseinilytica 10SCRN4D.* Fig 3 represents the hierarchical taxonomical classification of the strain. S1 Table shows the output result of the taxonomical identification when the *H. elongata* had the highest score of association.

Table 1. Sequence quality assessment report.

| Category | Report | Value | Expected value | Interpretation |
|---|---|---|---|---|
| **Read Quality (NanoPlot)** | % Reads > Q10 | 100% | 90% basecall accuracy – 10 | 100% of reads are good quality |
| | % Reads >Q12 | 44.8% | – | Nearly half the reads are over Q12 |
| | %Reads>Q15 | 2.0% | – | Few reads are over Q15 |
| **Assembly Statistics** | Number of contigs | 6 | Less than 10 | A low number of contigs indicates excellent contiguity |
| | # N's per 100 kbp | 0.00 | 0 | No gaps in the assembly, indicating fully resolved sequences |
| | GC content | 63.04% | 15% − 75% | Normal GC content range |
| | L50 | 2 | 2-10 | Only 2 contigs cover 50% of the total genome showing excellent assembly quality |
| | N50 | 890802 | – | highly contiguous assembly |
| **Overall Completeness** | Contigs > 50 kbp | 6 | >10 | All contigs are longer than 50 kbp, indicating high-quality assembly |

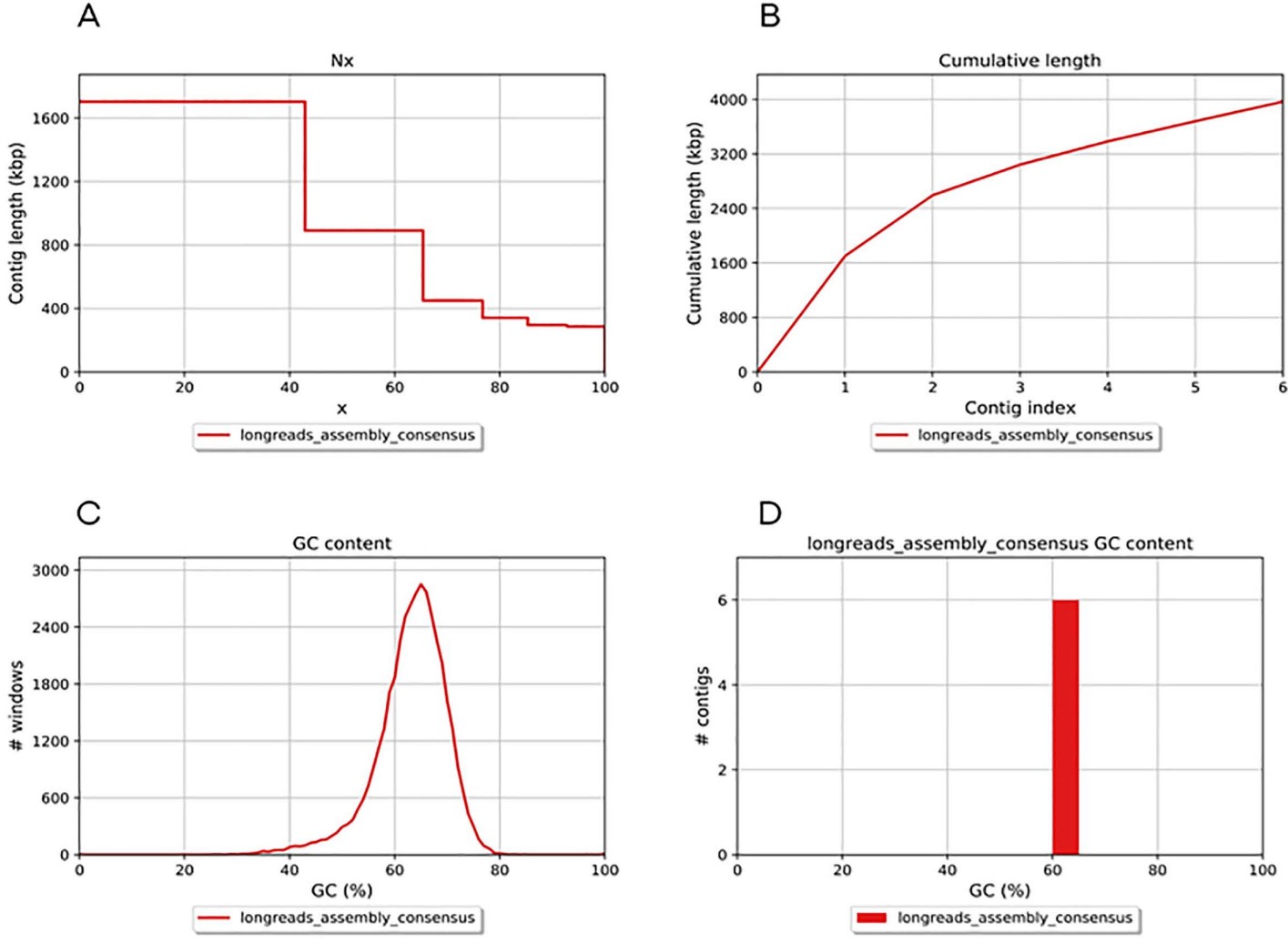

**Fig 1. Evaluation of quality of the genome assembly. A.** An Nx plot showing the assembly continuity and indicating a high-quality genome assembly with significant coverage achieved by large contigs. **B.** The plot shows the cumulative length of contigs from genome assembly as a function of contig index. A steep initial slope, which then levels off, indicates that a few long contigs make up a substantial part of the genome. **C.** The plot shows distribution of GC content across different windows of the assembled genome having a predominant and consistent GC content around 60%. **D.** The plot shows the peak of GC content across the contigs in the genome assembly and implies a uniform GC content of a little over 60%.

## Gene prediction and functional annotation

PROKKA predicted a total of 4308 genes within the genome. These genes were categorized as follows: 4227 genes annotated as conserved domain sequences (CDS), 12 genes annotated as rRNA, 68 genes annotated as tRNA and 1 gene annotated as tmRNA. Additionally, 1659 CDS were annotated as hypothetical proteins, representing genes with unidentified or uncertain functions. To further characterize these hypothetical proteins, they were subjected to analysis using Conserved Domain Database (CDD), where 737 hypothetical proteins were identified as specific proteins with defined functions and 396 hypothetical proteins were linked to their respective superfamilies, providing functional insights. However, 526 hypothetical proteins remained uncharacterized, representing sequences with no detectable matches to known proteins or superfamilies (S1–S4 Figs).

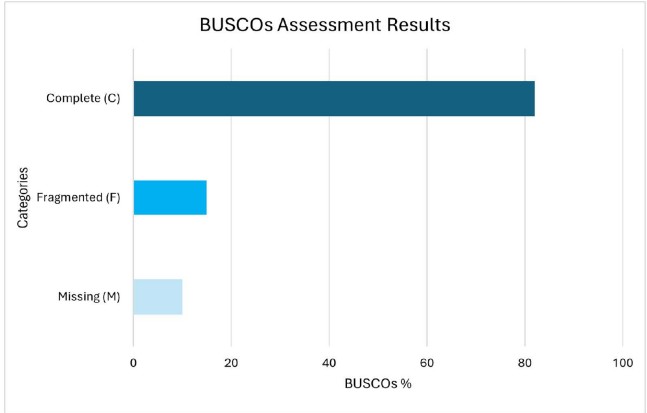

**Fig 2. BUSCO assessment results displaying three categories of genomic completeness.** Complete **(C)**, Fragmented **(F)**, and Missing **(M)**. The Complete (C) category dominates with approximately 81% of BUSCOs, including 505 single-copy and 1 duplicated. The Fragmented (F) category accounts for about 12%, while the Missing (M) category represents roughly 7%. The chart highlights the high quality and completeness of the genome assembly.

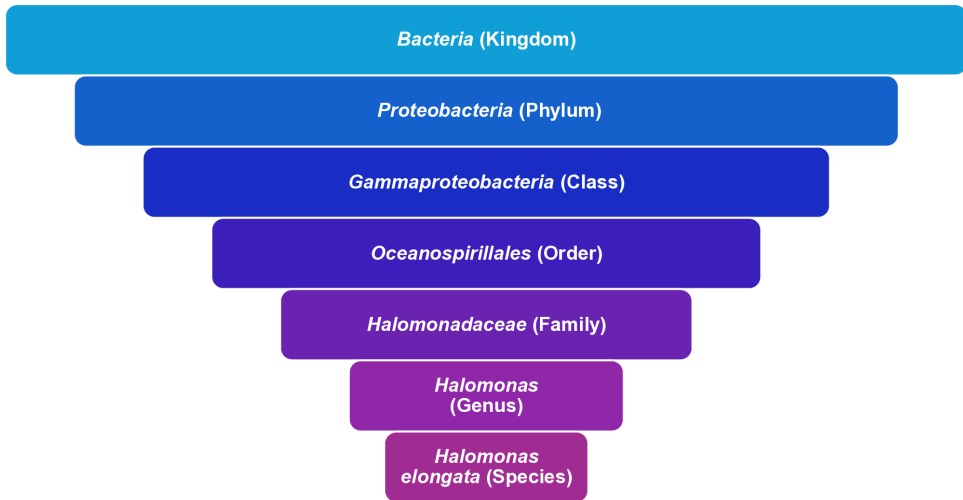

**Fig 3. Taxonomical classification of Halomonas elongata.**

In addition to PROKKA, PRODIGAL and GeneMarkS2 were utilized for gene prediction (Table 2). PRODIGAL predicted 4234 genes, of which 3785 genes were annotated using EggNOG-mapper. GeneMarkS2 predicted 4280 genes, with 3861 genes annotated via EggNOG-mapper. Gene ontology (GO) assignments were carried out using InterProScan and Blast2GO. The sequence distribution based on Biological, Cellular, and Molecular functions is summarized in Fig 4. 100 proteins were categorized under GO:0006805, corresponding to xenobiotic degradation, indicating potential involvement in detoxification processes. Notably, no proteins were categorized under GO:0019439, which corresponds to aromatic compound catabolism, highlighting a lack of direct annotations related to this specific function.

**Table 2. Comparison of the gene prediction and annotation results.**

| Gene Prediction and Annotation Tool | Total length | No. of genes predicted | No. of genes annotated | No. of CDS | No. of rRNAs | No. of tRNAs | No. of Hypothetical Proteins | No. of predicted genes not annotated |
|---|---|---|---|---|---|---|---|---|
| PROKKA | 1152142 | 4308 | 4308 | 4227 | 12 | 68 | 1659 | – |
| PRODIGAL+ EggNOG-mapper | 1157291 | 4234 | 3785 | 3657 | – | – | 128 | 449 |
| GenemarkS2+ EggNOG-mapper | 1161458 | 4280 | 3861 | 3732 | – | – | 129 | 419 |

"Total length" refers to the total sequence length (in base pairs) that was processed by each tool to be annoatable; "No. of genes predicted" includes all coding and non-coding sequences identified by the gene prediction tool; "No. of genes annotated" includes only those with functional annotation assigned by the annotation tool; "CDS" refers to protein-coding sequences among the annotated genes; "Hypothetical proteins" are predicted proteins without functional annotation among the predicted genes; "Predicted genes not annotated" indicates sequences that were predicted to be genes by the gene prediction tool but were not annotated by the annotation tools.

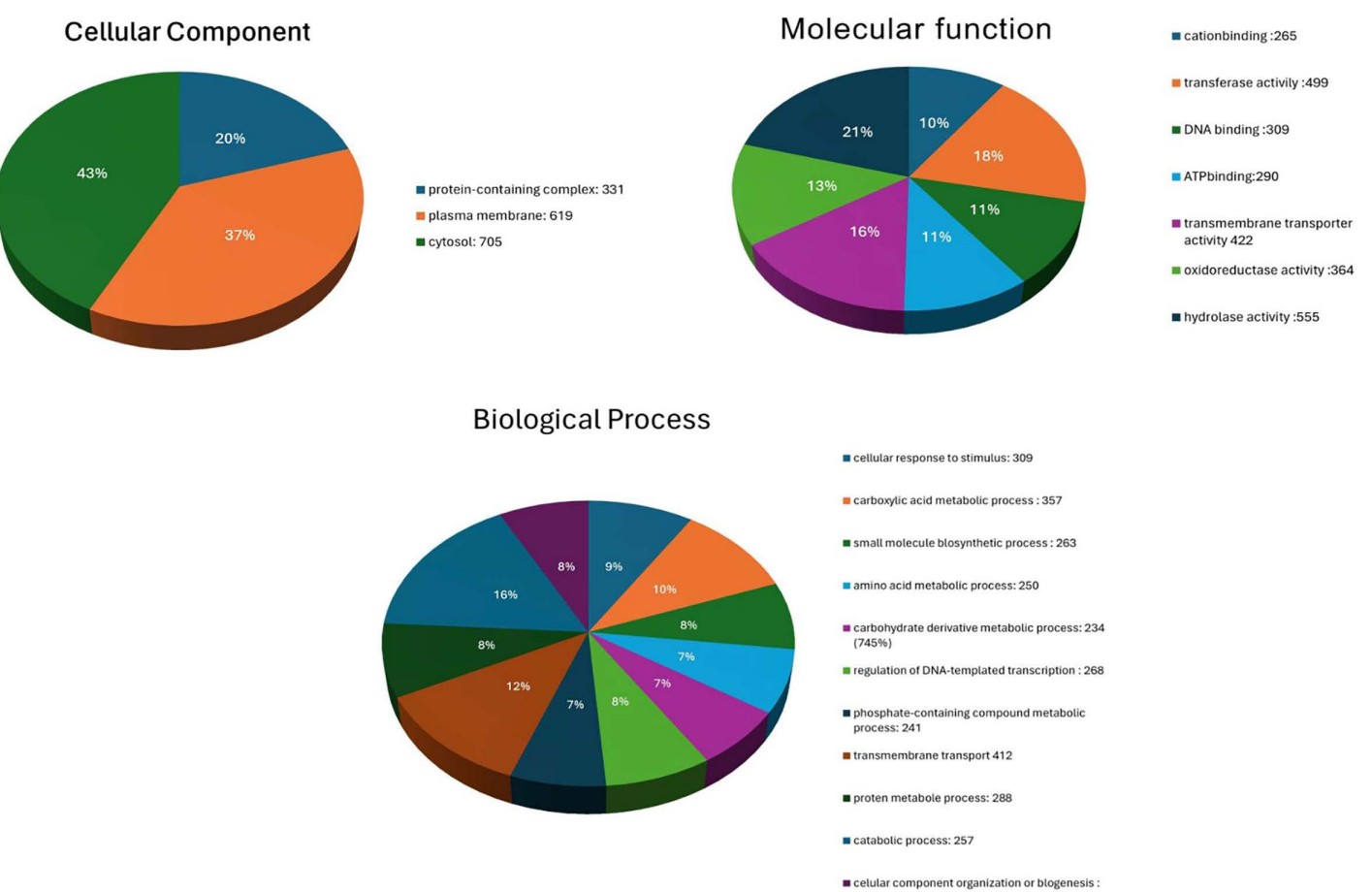

**Fig 4. GO analysis provides a functional snapshot of the strain's genomic capacity.** The image displays GO results distributed across three major categories: Cellular Component, Molecular Function, and Biological Process. Cytosolic and membrane-associated proteins suggest critical metabolic and transport-related roles; Enzymes involved in transferase, hydrolase, and oxidoreductase activities support the degradation of complex organic compounds like coronene; The biological processes further underline the bacterium's capacity to adapt, organize its cellular machinery and perform specialized functions.

## Identification of genomic features

Operon Mapper identified 2013 operons out of which at least 9 were associated with aromatic compounds degradation. Two CRISPR sites were identified, one in utg000001l contig and the other in utg000005l contig (S2 Table). No *cas* sites were detected.

In total, 436 repeat regions were identified. These regions mainly comprised of simple repeats. In addition to the simple repeats, LINEs, SINEs, rRNA and tRNA repeats were also detected (Fig 5). 1288 Palindrome, 121 mobile elements and 47 Horizontal transfer genes were also found. antiSMASH was able to identify three secondary metabolite regions, namely, ectoine, NRPS/ NRPS metallophore, RiPP like protein. Fig 6 shows the gene clusters of the three secondary metabolite biosynthesis. CARD [49] detected 3 antibiotic resistance genes, namely, adeF, rsmA and qacG. Fig 7 represents the whole genome of the bacteria created using Proksee web-based tool [59].

## Ectoine production

In recent years, ectoine has been extensively studied for commercial application due to its ability to stabilize cellular components such as DNAs and proteins [60]. From the annotation results, 3 of the enzymes required for ectoine synthesis, namely, Diaminobutyric acid acetyltransferase (ectA), L-2,4-diaminobutyrate-2-oxoglutarate transaminase (ectB) and Ectoine synthase (ectC) were identified. Additionally, Ectoine hydroxylase (ectD) involved in the conversion of ectoine to 5-hydroxyectoin was also found. EctD is not commonly found in all ectoine biosynthesizing bacteria. 5-hydroxyectoin has superior stress-relieving properties [61].

## Functional and pathway analysis

The RAST analysis revealed that only 32% of the genome was associated with subsystem categories. Overall, 4393 coding sequences in the genome were Identified using RAST. Of these, 1398 coding sequences were linked to one or more subsystems in the database. Within the category of aromatic compound metabolism, 30 features/genes were identified, highlighting the organism's potential role in degrading aromatic compounds. Additionally, 2 genes categorized under miscellaneous subsystems were associated with aromatic dioxygenase activity (S3 Table). RAST also identified other important subcategories including resistance to antibiotics and toxic compounds, Invasion and intracellular resistance, Prophage and phage packaging machinery suggesting mechanisms for survival in challenging environments, host interaction capabilities and phage related function (Fig 8).

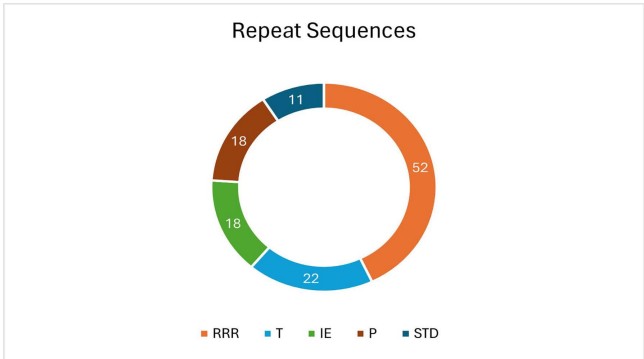

**Fig 5. Repeat sequences found in the genome.** T-transfer DNA, IE- Integration/excision, RRR-Replication/Recombination/Repair, P- Phage, STD- Stability/Transfer/Defense.

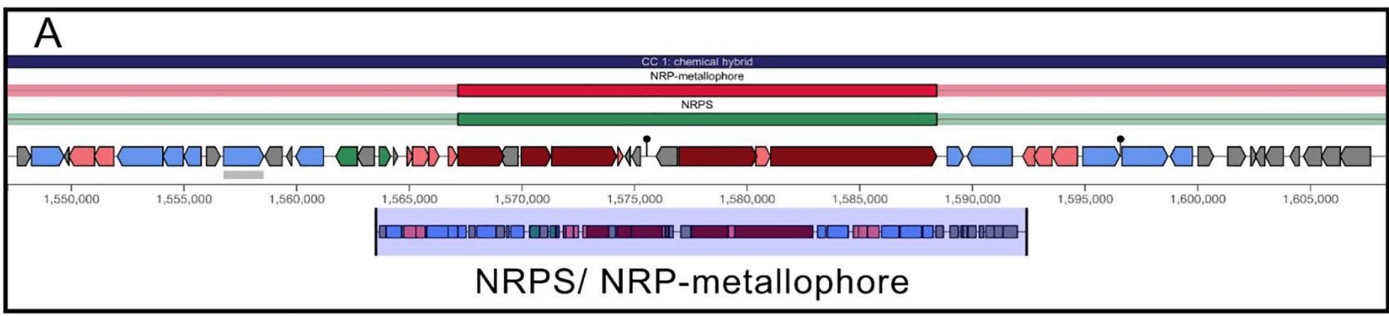

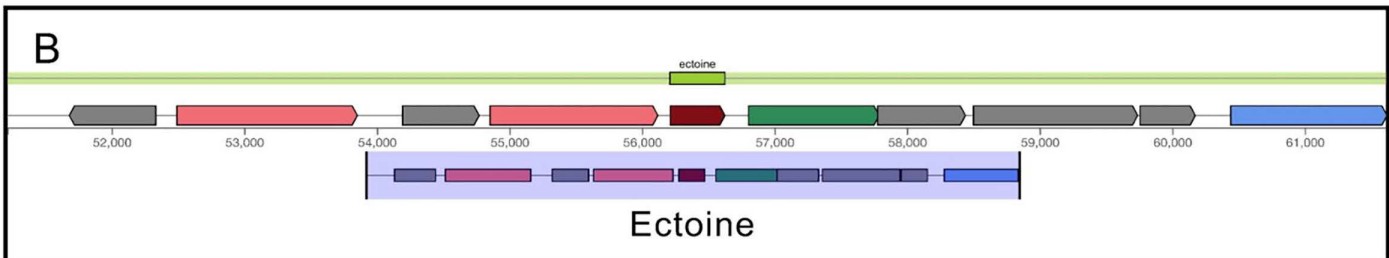

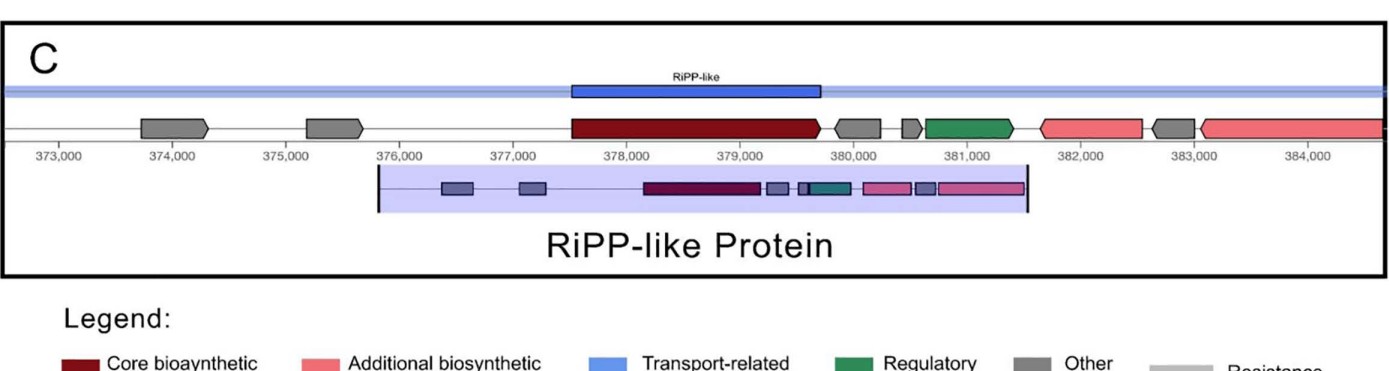

Legend:

| | | | | | |
|---|---|---|---|---|---|
| ■ Core biosynthetic genes | ■ Additional biosynthetic genes | ■ Transport-related genes | ■ Regulatory genes | ■ Other genes | ▬ Resistance |

**Fig 6. Secondary metabolite clusters of NRPS/NRP-metallophore, ectoine and RiPP-like protein.** Each gene clusters consist of core and additional biosynthetic genes, regulatory genes, transport-related genes and resistance gene.

Pathway analysis using InterProScan results identified 266 KEGG pathways in addition to 1866 sequences that were found to be associated to one or more pathways. KASS produced KO list which helped in mapping the pathways. A total of 2101 genes were annotated with KO numbers. Pathway mapping allowed to see the different pathways our bacterial genome aligns with in the KEGG databases. Under the category of xenobiotic degradation pathways, 13 partial KEGG pathways were identified. One reason to explain this would be the probable presence of alternative pathways which may not be a part of the standard KEGG modules. S5 Fig shows the pathway map for the degradation of PAHs mediated by cytochrome P450.

## Discussion

The advent of LRST, such as ONS used in our study, has revolutionized microbial genomics by enabling high-contiguity assemblies and the resolution of complex genomic features. This study leverages the strengths of LRST to elucidate the genomic adaptations of *H. elongata (*previously *H. caseinilytica 10SCRN4D*). Transitioning from 16S rRNA sequencing to

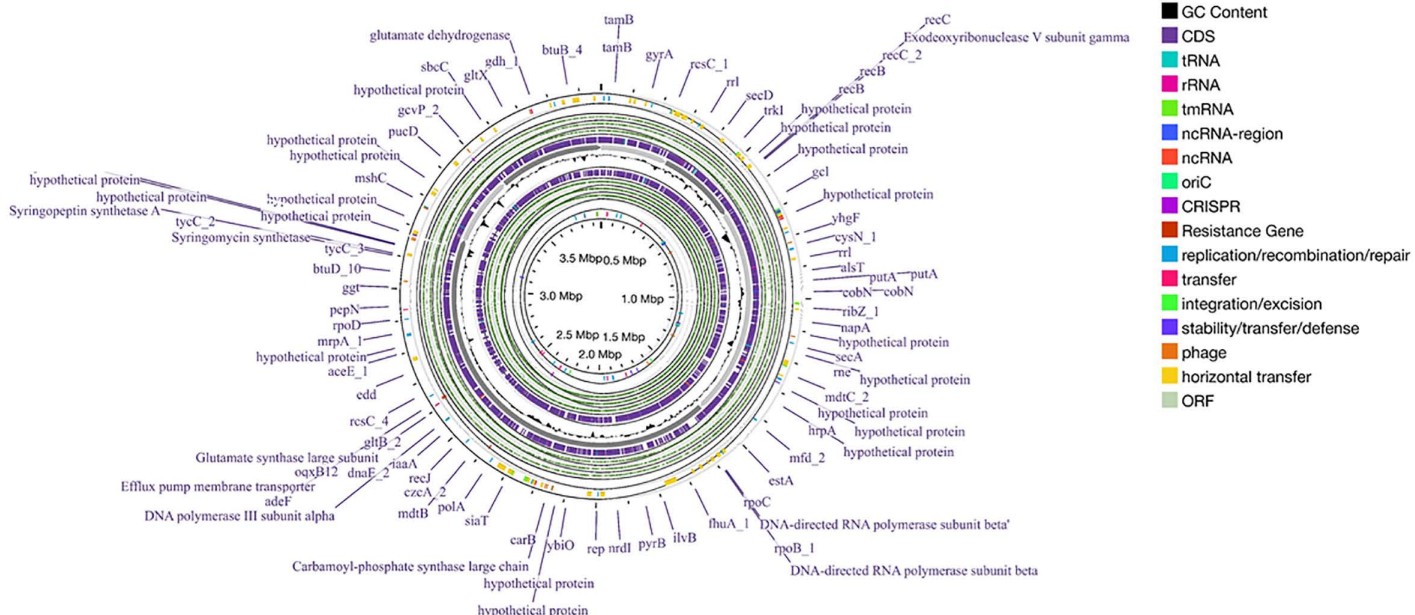

**Fig 7. Schematic circular map of the whole genome of Halomonas elongata.** The map illustrates the location of the different genetic components of the bacterial genome. Used under creative commons license.

whole genome sequencing resulted in the reclassification of our strain from *H. caseinilytica 10SCRN4D to H. elongata*, accentuating the evolving nature of bacterial taxonomy and the limitations of 16S rRNA-based identification methods [62]. This phenomenon is not unique to our study, as similar reclassifications have been observed in other bacterial genera. For instance, WGS-based analyses led to the proposed combination of two *Clostridium* species in a 2021 study [63], and another research effort reclassified an *Elizabethkingia miricola* strain as *E. bruuniana* [64]. These recurring instances of species reclassification can be attributed to the insufficient resolution of 16S rRNA-based identification, particularly when distinguishing closely related species within genetically complex genera like *Halomonas* [65]. The genetic similarity among *Halomonas* species complicates accurate classification when relying solely on the 16S rRNA gene [62]. Relatively, LRST provides a comprehensive genetic landscape, enabling more precise and nuanced species determination. Our study's findings highlight the advantages of LRST in uncovering subtle genomic differences crucial for accurate taxonomic classification [64].

Recent studies have highlighted the potential of halophilic bacteria in degrading HMW-PAHs under saline conditions. *Nanca et al*. [66] isolated halophilic bacteria from Philippine salt beds capable of degrading pyrene, fluorene, and fluoranthene, demonstrating the versatility of halophiles in PAH degradation. Other studies have demonstrated the ability of various *Halomonas sp*. to degrade aromatic hydrocarbons under hypersaline conditions. For instance, *H. organivorans* has been reported to degrade phenol, salicylate, and benzoate, utilizing pathways involving phenol hydroxylase and catechol 2,3-dioxygenase enzymes [67].Similarly, *Halomonas* sp. strain ML-15 was shown to degrade phenanthrene effectively under haloalkaliphilic conditions, emphasizing the adaptability of *Halomonas* species to extreme environments [68]. *Halomonas sp.* strain C2SS100 has exhibited the capacity to degrade hydrocarbons under high salinity, highlighting the genus's adaptability to extreme environments [69].Our strain of study, as observed from our previous research, was capable of degrading coronene at the same rate as that of any LMW-PAHs and at a salinity ranging between 0.5% to 10% [18]. Renowned for their ectoine producing ability, *H. elongata* is a halophilic γ-proteobacterium that has an optimal growth at salt concentrations ranging from 3.5% to 20% NaCl [62]. Despite the *Halomonas sp.* remarkable capability, the

## Subsystem Category Distribution

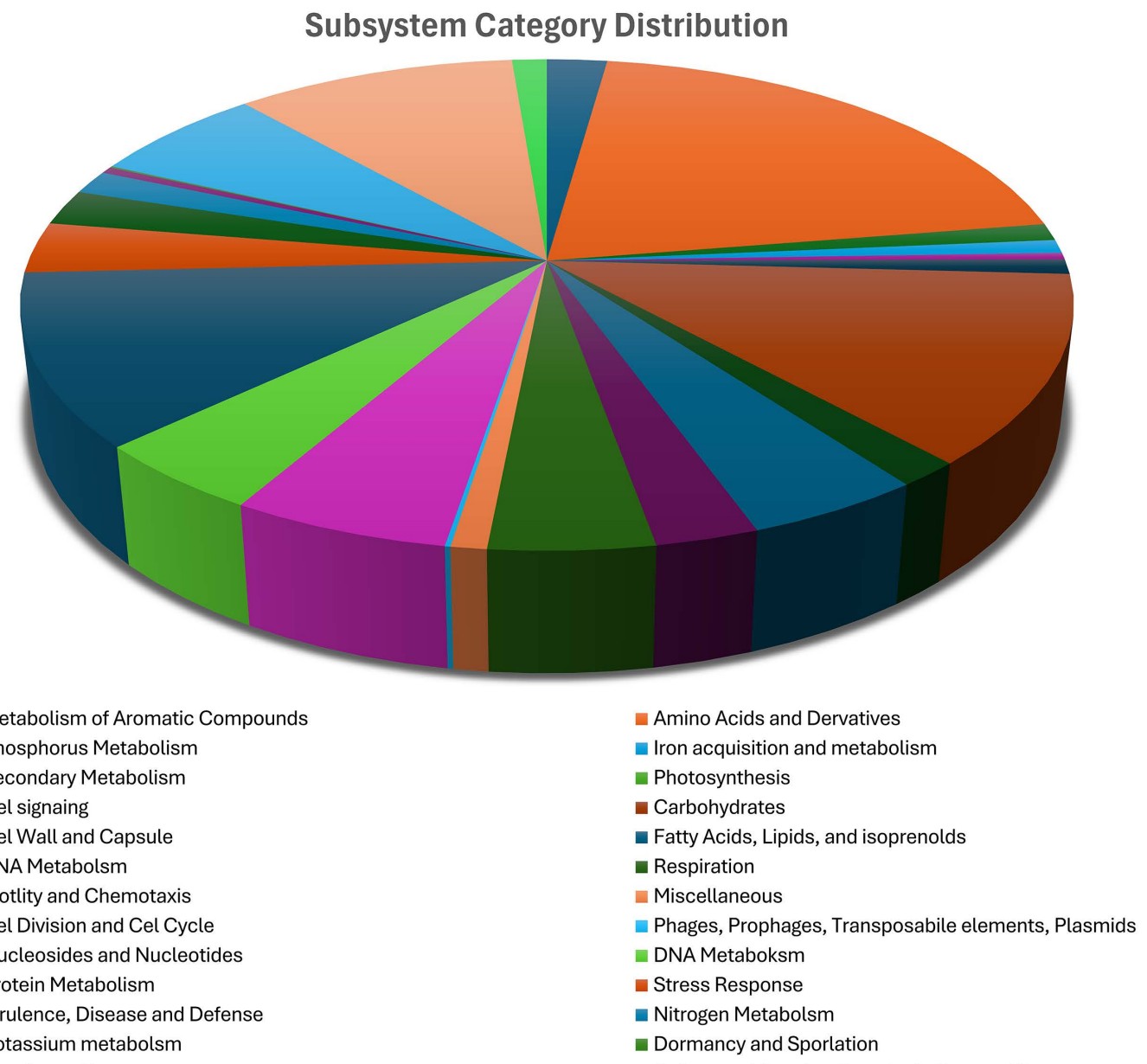

**Legend (left column):**
- ■ Metabolism of Aromatic Compounds
- ■ Phosphorus Metabolism
- ■ Secondary Metabolism
- ■ Cel signaing
- ■ Cel Wall and Capsule
- ■ RNA Metabolsm
- ■ Motlity and Chemotaxis
- ■ Cel Division and Cel Cycle
- ■ Nucleosides and Nucleotides
- ■ Protein Metabolism
- ■ Virulence, Disease and Defense
- ■ Potassium metabolsm
- ■ Membrane Transport
- ■ Sulfur Metabolism

**Legend (right column):**
- ■ Amino Acids and Dervatives
- ■ Iron acquisition and metabolism
- ■ Photosynthesis
- ■ Carbohydrates
- ■ Fatty Acids, Lipids, and isoprenolds
- ■ Respiration
- ■ Miscellaneous
- ■ Phages, Prophages, Transposabile elements, Plasmids
- ■ DNA Metaboksm
- ■ Stress Response
- ■ Nitrogen Metabolsm
- ■ Dormancy and Sporlation
- ■ Cofactors, Vitamins, Prosthetic Groups, Pigments

**Fig 8. A pie chart representing the subsystems coverage of the genome using the RAST database.** Subsystems include "Metabolism of Aromatic Compounds," "Amino Acids and Derivatives," "Carbohydrates," "Phosphorus Metabolism," "Secondary Metabolism," "Stress Response," "Membrane Transport," and others. The chart highlights functional diversity with significant portions for metabolic, transport, and stress adaptation-related processes.

degradation of HMW PAHs such as coronene by *H. elongata* has not been previously reported in the literature, highlighting the novelty and significance of our findings.

The gene prediction and annotation results from multiple tools (PROKKA, PRODIGAL, and GenemarkS2) provide a comprehensive view of the *H. elongata* strain's genomic content. The consistent gene count across different prediction

algorithms (4308, 4234, and 4280, respectively) lends credibility to the overall gene density and supports the robustness of the genome assembly. PROKKA's annotation revealed a high proportion of protein-coding sequences (4227 CDS) and essential RNA genes, indicating a complete set of translational machinery crucial for cellular function [70]. However, high number of hypothetical proteins (1659 out of 4227 CDS) initially annotated by PROKKA highlights the current limitations in our knowledge of bacterial gene functions, particularly in less-studied genera like *Halomonas*. The subsequent analysis of these hypothetical proteins using CDD reduced the number of truly uncharacterized proteins from 1659 to 526. This significant reduction emphasizes the importance of using multiple annotation tools and databases to maximize functional assignments as done in this study. The remaining 526 hypothetical proteins with no identified domains or superfamilies represent potential targets for future experimental characterization. These could be genes unique to *Halomonas* or even strain-specific adaptations, possibly playing roles in the organism's specific environmental niche, and in our case, the ability to degrade coronene [65,71].

Comparative genomic analyses have further elucidated the mechanisms underlying PAH degradation in halophilic bacteria. *Pontibacillus chungwhensis* HN14, for example, possesses gene clusters associated with PAH degradation pathways, emphasizing the genetic basis for their catabolic capabilities [72]. These findings align with our genomic analysis of *H. elongata*, which revealed genes involved in aromatic compound degradation, antibiotic resistance, and stress adaptation. GO analysis with InterProScan and Blast2GO provided a general overview of the genome's functional landscape. The identification of 100 proteins categorized under xenobiotic metabolic processes (GO:0006805) potentially addresses the strain's ability to degrade PAHs. However, the absence of proteins categorized under aromatic compound catabolism (GO:0019439) presents a contradiction that could be interpreted in several ways, including the possibility of alternative or novel pathways for aromatic compound degradation not yet captured by current GO terms or specific genes may not be well-represented in existing databases [62,73]. This can be backed by the knowledge that GO has not completely established its ontology and has limited coverage of multi-functional genes [73].

RAST and KEGG pathway analyses provide insights into the strain's functional capabilities and metabolic potential. The relatively low percentage of genes assigned to RAST subsystems (32%) implies a substantial number of unique or poorly characterized genes [56]. The identification of 13 partial KEGG pathways related to xenobiotic degradation is of interest, although the absence of complete pathways could be due to the use of unique or modified pathways not defined in KEGG modules or the genes may have slight variations resulting in them not being assigned with a KO number [74].

The results from Operon Mapper, CRISPR analysis, repeat region identification, CARD and antiSMASH provide valuable insights into the genomic organization and functional potential. Particularly, the nine operons associated with aromatic compound degradation corroborates the earlier Gene Ontology results indicating xenobiotic degradation potential, and at the same time suggesting that *H. elongata* strain under study may possess specialized pathways to break down aromatic compounds. The detection of CRISPR site but the absence of *cas* genes is intriguing. This either means the CRISPRs identified are non-functional, orphan CRISPR arrays or the *cas* genes are present but not identified by the current annotation methods [75]. Antibiotic resistance to fluoroquinolone, tetracycline, diaminopyrimidine and phenolic compounds is mainly due to the presence of efflux proteins rsmA and adeF. The gene qacG confers to it's resistance to disinfecting agents and antiseptics [49]. Additionally, the 47 genes found to be horizontally transferred could play a role in the strain's ability to degrade coronene. This assumption is based off of Han and co-workers research in 2025 where they observed that the ability of *Altererythrobacter* sp. H2 to degrade PAHs was due to horizontal gene transfer [76].

Secondary metabolite regions, particularly ectoine biosynthesis cluster is consistent with the halophilic nature of the strain as ectoine is used for osmotic balance in halophilic bacteria [60,77]. The presence of NRPS clusters, including one encoding a metallophore, would suggest the capacity to produce complex secondary metabolites participating in metal acquisition or other ecological interactions that plays a prominent role in bioremediation. RiPP-like (Ribosomally synthesized and Post-translationally modified Peptide) cluster signifies the potential for the production of bioactive peptides which has the prospect to be explored in anti-microbial activity studies [78]. Most importantly, the presence of 4 genes

involved in ectoine synthesis makes our strain an important candidate for research in cosmetics and medicine. On the other hand, RAST and KEGG pathway analyses provided insights into the strain's functional capabilities and metabolic potential. The relatively low percentage of genes assigned to RAST subsystems (32%) implies a substantial number of unique or poorly characterized genes [56]. The identification of 13 partial KEGG pathways related to xenobiotic degradation is of interest, although the absence of complete pathways could be due to the use of unique or modified pathways not defined in KEGG modules or the genes may have slight variations resulting in them not being assigned with a KO number [74].

While this study provides a detailed genomic analysis of *H. elongata* and its potential role in coronene degradation, it is constrained by the lack of functional validation through transcriptomic or proteomic data. The genes and pathways identified here, though computationally annotated, require experimental confirmation to establish their specific roles in PAH metabolism. Given the structural complexity and limited existing knowledge regarding the biodegradation pathways for coronene, we initially hypothesized that *H. elongata* might utilize established degradation pathways known for other PAHs, such as naphthalene or phenanthrene. Surprisingly, genome analysis revealed that *H. elongata* lacks key enzymes commonly associated with these canonical PAH degradation pathways. As the degradation intermediates of coronene were not characterized, our understanding of the complete metabolic pathway is limited. Future studies incorporating gene knockout, heterologous expression, and metabolite profiling will be essential to verify the function of key enzymes and to clarify the molecular mechanisms enabling coronene degradation under high salinity conditions.

## Conclusion

This study presents a comprehensive genomic analysis of *H. elongata* (previously classified as *H. caseinilytica*), revealing its exceptional potential for degrading coronene, a HMW-PAH, under saline conditions. By utilizing LRST coupled with advanced bioinformatics tools, we identified specific genetic components and pathways related to xenobiotic metabolism, production of secondary metabolites, and adaptive mechanisms such as horizontal gene transfer and CRISPR arrays. These genetic insights highlight the organism's adaptability and underscore its significant promise for environmental applications.

Broader implications of our findings include potential utilization of *H. elongata* in bioremediation strategies for marine and hypersaline ecosystems contaminated with complex hydrocarbons, as well as opportunities for industrial biotechnology applications, particularly involving halotolerant secondary metabolite production like ectoine. However, this genomic study faces limitations, notably the absence of functional validation through transcriptomic, proteomic, and metabolomic analyses. Consequently, the specific biochemical mechanisms underlying coronene degradation remain hypothetical and require confirmation through experimental studies.

Future research should explicitly focus on validating the identified metabolic pathways, characterizing unannotated or hypothetical proteins, and exploring industrially relevant secondary metabolites. Targeted genetic experiments, including gene knockouts and metabolite profiling, are essential next steps to fully harness the biotechnological and environmental potentials of *H. elongata*. Such future studies will significantly strengthen our understanding and enable the practical deployment of this microorganism to sustainably mitigate PAH contamination in challenging environmental settings.

## Supporting information

**S1 Fig. Genes associated with superfamilies predicted on CDD.**
(DOCX)

**S2 Fig. Genes associated with superfamilies predicted on CDD.**
(DOCX)

**S3 Fig. Genes associated with superfamilies predicted on CDD.**
(DOCX)

**S4 Fig. Genes associated with superfamilies predicted on CDD.**
(DOCX)

**S5 Fig. Pathway mapping of metabolism of xenobiotics by cytochrome P450 from KASS The green boxes indicate the genes that are present in our gene list while the blue boxes indicate those that are absent but should have been present.** Where EC: 2.5.1.18 is glutathione S-transferase.
(DOCX)

**S1 Table. Kraken2 taxonomical classification.**
(DOCX)

**S2 Table. CRISPR sites.**
(DOCX)

**S3 Table. RAST Annotation corresponding to aromatic compound metabolism.**
(DOCX)

## Author contributions

**Conceptualization:** Alexis Nzila.

**Formal analysis:** Thasneema Rafic, Omer Salem Alkhnbashi, Karthikeyan G.

**Funding acquisition:** Alexis Nzila.

**Investigation:** Alexis Nzila.

**Methodology:** Thasneema Rafic, Mohammed Alarawi, Assad Al-Thukair, Ajibola H. Okeyode.

**Project administration:** Alexis Nzila.

**Resources:** Omer Salem Alkhnbashi.

**Software:** Thasneema Rafic, Mohammed Alarawi, Omer Salem Alkhnbashi.

**Supervision:** Omer Salem Alkhnbashi.

**Writing – original draft:** Thasneema Rafic, Karthikeyan G, Alexis Nzila.

**Writing – review & editing:** Thasneema Rafic, Omer Salem Alkhnbashi, Assad Al-Thukair, Ajibola H. Okeyode, Karthikeyan G, Alexis Nzila.

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
