## [Decision Letter · Decision Letter 0]

15 Apr 2025

Dear Dr. Nzila,

Thank you for submitting your manuscript to PLOS ONE. After careful consideration, we feel that it has merit but does not fully meet PLOS ONE’s publication criteria as it currently stands. Therefore, we invite you to submit a revised version of the manuscript that addresses the points raised during the review process.

We look forward to receiving your revised manuscript.

Kind regards,

Bijay Kumar Behera, Ph.D.

Academic Editor

PLOS ONE

 [This study was supported by project number INMW2301 by the Interdisciplinary Research Center for Membranes and Water Security (IRC-MWS), King Fahd University of Petroleum and Minerals (KFUPM) Dhahran, Saudi Arabia.].

3. In the online submission form, you indicated that [All the data are available upon request].

Additional Editor Comments:

The manuscript entitled "Whole Genome sequencing, Characterization and Analysis of coronene degrading Bacterial strain Halomonas elongata" needs major revision. The two reviewers recommended revising the manuscript.

Reviewers' comments:

Reviewer's Responses to Questions

**Comments to the Author**

1. Is the manuscript technically sound, and do the data support the conclusions?

Reviewer #1: Yes

Reviewer #2: Partly

2. Has the statistical analysis been performed appropriately and rigorously?

Reviewer #1: Yes

Reviewer #2: N/A

3. Have the authors made all data underlying the findings in their manuscript fully available?

Reviewer #1: No

Reviewer #2: Yes

4. Is the manuscript presented in an intelligible fashion and written in standard English?

Reviewer #1: Yes

Reviewer #2: Yes

Reviewer #1: The study addresses an important environmental issue—bioremediation of PAHs, which are persistent organic pollutants. This manuscript investigates the whole-genome sequencing and characterization of Halomonas elongata, a bacterial strain capable of degrading coronene, a high-molecular-weight polycyclic aromatic hydrocarbon (PAH). The study highlights the biotechnological potential of Halomonas elongata in bioremediation applications. The manuscript is well-structured and presents valuable genomic insights into Halomonas elongata. However, the following improvements in the manuscript would enhance its impact.

1. The abstract should provide a more concise yet informative summary, including key findings and their implications. Here, KEGG KAAS can be written as KAAS only as first K in KAAS stands for KEGG.

2. The introduction needs a stronger rationale explaining why Halomonas elongata is particularly significant for coronene degradation compared to other bacteria. Key studies on bacterial PAH degradation should be included.

3. More details on sequencing technology, genome assembly tools, and annotation methods would enhance reproducibility. Further, details on collection of sample and isolation of bacteria in in the beginning of materials and method are required. Name of the sequencing technology should be mentioned at line no 110 in full form

4. The study aims to provide insights into the genetic mechanisms underlying coronene degradation. What molecular mechanism you unraveled for coronene degradation by genome sequencing of H elongata should be elaborated and may be presented using an appropriately designed flow chart.

5. What is the basis of the cutoff 1e-06 in line 126?

6. Line no. 154 k-value=1 should be justified.

7. Table 1 should be properly placed for easy understanding, statistics related to Q20/Q30 should be given as minimum Q20 is an widely accepted Phred Score. The N50 value should also be provided in this table.

8. Table 2 is difficult to follow. Check the figures carefully. What is total length in Table 2. Sufficient information needs to be provided for each table and figures for making them self-explanatory.

9. Most of the figures are of poor quality having very small fonts. The clarity of figures and tables should be ensured—some may need better labelling or explanations. Figure 5 may be represented as a 3d bar chart.

10. Experimental validation of identified genes (e.g., knockout studies) would strengthen claims about metabolic pathways.

11. The discussion should include a more thorough comparison with previous studies on Halomonas elongata and other hydrocarbon-degrading bacteria. Functional analysis of specific genes responsible for coronene degradation should be elaborated. Metabolic pathways should be discussed, with a visual representation would improve comprehension.

12. The conclusion should emphasize the broader implications, such as potential applications in bioremediation and industrial use. Limitations and future research directions should be explicitly stated.

Minor Comments

1. KEGG KAAS can be written as KAAS only

2. LRS, CDD, KO, BUSCO, CRISPR, CARD etc. are to be expanded at their first instance.

3. Species name, and the words like in silico de novo etc. should be represented in italics.

4. Some phrases are repeatedly used,

5. In line 98. “Koren et al. in 2013” should be written as “Koren et al. [21]” and citation [21] at the end of the sentence should be removed.

6. Line: 219 Check carefully “Error! Reference not found..”

7. Improper capitalization of first letter in many words throughout this manuscript.

8. Some places connectivity of sentences is missing

9. Line no 271: “aligns with from..”.. aligns with what?

Reviewer #2: General comments

1. Introduction needs some more expansion specially for the recent work published worldwide.

2. Where bacteria collected from? And full procedure regarding etc.

3. Background of the bacteria lacking from the manuscript.

4. Detailed procedure needed about whole genome sequencing and quality assessment.

5. Some headings/ subheading / paragraphs missing the major citation, please update accordingly.

6. There are several capitalizations between the sentence and heading. Authors should go throughout the manuscript for betterment of the article such as tile of fig 3, table 2 etc.

7. Discussion part needs more expansion especially relation with the recent work published.

8. References needs for formatting according the journal guidelines, there are several mistakes such as reference no 10 in the list. Fermentation 2022, Vol 8, 412 Page 260 2022;8:260.

9. Clarity of the figures blurry, not even able to understand the writeup materials.

**Do you want your identity to be public for this peer review?** For information about this choice, including consent withdrawal, please see our Privacy Policy

Reviewer #1: **Yes: ** Dr. Tanmaya Kumar Sahu

Reviewer #2: No

---

## [Author Response · Author response to Decision Letter 1]

19 Jun 2025

RESPONSE TO THE REVIEWERS’ COMMENTs PONE-D-25-12285

First of all, we thank the reviewers for their valuable time in reviewing manuscripts. We have addressed their reviewer’s comment, and enclosed below is the point-by-point response to these comments.

Reviewer #1

1. The abstract should provide a more concise yet informative summary, including key findings and their implications.

Our response:

The abstract has been revised, and it is more concise and informative.

2. The introduction needs a stronger rationale explaining why Halomonas elongata is particularly significant for coronene degradation compared to other bacteria. Key studies on bacterial PAH degradation should be included.

Our response:

In light of the reviewer’s comments, we have expanded the introduction to incorporate new information on the degradation of PAHs in particular and that of the coronene in general. In addition, we listed around appropriate references that could help the readers to have a broader view of this topic.

Regarding the bacterial strain, it has been isolated as part of a different study,, and we have quoted this reference (Okeyode et al. 2023). However, in light of the reviewer comments (the same point has also been raised by reviewer 2), we have added the main information on the characteristics of this strain in the “Introduction section. All detailed information regarding this strain is summarised in Okeyode et al. 2003).

3. More details on sequencing technology, genome assembly tools, and annotation methods would enhance reproducibility. Further, details on collection of sample and isolation of bacteria in in the beginning of materials and method are required. Name of the sequencing technology should be mentioned at line no 110 in full form

Our response:

• We have addressed the comments related to Bioinformatics in the “Material & Method Section”.

• However, regarding the sources of samples and isolations of bacteria, as discussed in the previous section, this bacterial strain were isolated as part of a previous work, and the we have summarised the characteristics of this strain in the “Introduction”, and have quoted the reference linked to this work. We cannot add this information in the Material Methods since it is not part of the current work.

4. The study aims to provide insights into the genetic mechanisms underlying coronene degradation. What molecular mechanism you unraveled for coronene degradation by genome sequencing of H elongata should be elaborated and may be presented using an appropriately designed flow chart.

5. Our response:

Coronene is one of the most structurally complex PAHs, and currently, there is limited knowledge about the specific molecular mechanisms involved in its biodegradation. Initially, we hypothesized that the degradation of coronene might follow similar pathways to those well-characterized for other PAHs, such as naphthalene, phenanthrene, or benzo[a]pyrene. However, upon detailed genomic analysis of Halomonas elongata, no key enzymes typically involved in known PAH degradation pathways were found, from the annotated genome.

This unexpected result strongly suggests that H. elongata likely employs an alternative, previously undocumented pathway for coronene degradation. Identifying and confirming this novel route will require further experimental investigation, including metabolomic analyses to characterize intermediate degradation products. Only through such comprehensive metabolic profiling can we accurately delineate the complete biochemical pathway. These points have been highlighted in the last paragraph of the “Discussion Section.

6. What is the basis of the cutoff 1e-06 in line 126?

7. Our response:

The E-value cutoff of 1e-06 was chosen to ensure that only highly significant matches were retained during gene prediction. This stringent threshold minimizes false positives while maintaining sensitivity to detect true homologs, improving the accuracy and reliability of functional annotations.

8. Line no. 154 k-value=1 should be justified.

Our response:

A k-value of 1 was used in MobileOG-DB to ensure that even single, high-confidence matches to known mobile element proteins were detected, which is especially important for identifying potentially novel or incomplete mobile elements in environmental genomes.

9. Table 1 should be properly placed for easy understanding, statistics related to Q20/Q30 should be given as minimum Q20 is an widely accepted Phred Score. The N50 value should also be provided in this table.

Our response:

• Table location is based on the journal requirement. Thus, we wisht to stick to the journal instruction, unless the Editor states otherwise..

• Since this study used Oxford Nanopore long-read sequencing, traditional Phred-based Q20/Q30 scores—commonly reported in short-read technologies like Illumina—are not directly applicable. Nanopore sequencing evaluates read quality using different metrics (such as mean read quality scores, read length distributions, and base-calling accuracy), which are reported separately in our quality assessment.

10. Table 2 is difficult to follow. Check the figures carefully. What is total length in Table 2. Sufficient information needs to be provided for each table and figures for making them self-explanatory.

Our response:

We have carefully revised Table 2 to improve clarity, layout, and readability.

• The column “Total length” has now been clearly labeled and explained in the table legend. It refers to the cumulative length (in base pairs) of the sequences analysed for gene prediction by each respective tool.

• We have ensured that each tool’s name is clearly and consistently labeled (e.g., “PRODIGAL + EggNOG-mapper” and “GeneMarkS2 + EggNOG-mapper” presented on a single line).

• A comprehensive table legend has been added to explain each column, including:

11. Most of the figures are of poor quality having very small fonts. The clarity of figures and tables should be ensured—some may need better labelling or explanations. Figure 5 may be represented as a 3d bar chart.

12. Our response:

• All figures have been regenerated or replaced using high-resolution formats to ensure improved clarity in both digital and print formats.

• Fig 5 has been converted to a bar graph.

• To counter the resolution problem related to fig 7, it has been submitted in PNG

13. Experimental validation of identified genes (e.g., knockout studies) would strengthen claims about metabolic pathways.

14. Our response:

We fully agree that experimental validation—particularly through gene knockout studies—would greatly enhance the strength of our claims regarding the metabolic pathways involved in coronene degradation. However, we wish to clarify that this study was designed as a computational genomic investigation to identify potential genes and pathways implicated in high-molecular-weight PAH biodegradation using long-read whole genome sequencing and advanced bioinformatics tools. Importantly, no known pathways for coronene degradation have been characterized to date. Therefore, our current approach was guided by comparative genomic analysis, assuming similarity to other PAH-degradation mechanisms. Interestingly, genes traditionally associated with PAH degradation (e.g., dioxygenases involved in pyrene or benzo[a]pyrene metabolism) were not found in our genome annotation, suggesting the existence of a novel or alternative degradation mechanism in Halomonas elongata. This point have been highlighted in the final paragraph of “Discussion section”

11. The discussion should include a more thorough comparison with previous studies on Halomonas elongata and other hydrocarbon-degrading bacteria. Functional analysis of specific genes responsible for coronene degradation should be elaborated. Metabolic pathways should be discussed, with a visual representation would improve comprehension.

Our response:

• We have added a detailed comparison of our findings with previous research on other hydrocarbon-degrading Halomonas species, including H. organivorans, Halomonas sp. ML-15, and Halomonas sp. C2SS100. These comparisons highlight both the shared capabilities and the distinct genomic features of H. elongata 10SCRN4D, particularly with respect to its coronene degradation potential under hypersaline conditions.

• We elaborated on the annotation results by highlighting the presence of genes associated with xenobiotic metabolism and the absence of canonical PAH-degradation enzymes (e.g., aromatic ring-hydroxylating dioxygenases). We discussed the possible roles of monooxygenases, hydrolases, and dehydrogenases found in the genome, and proposed that H. elongata may employ an alternative or novel degradation route. The implications of these genes, along with the remaining 526 uncharacterized proteins, are discussed as promising targets for future experimental work.

• We agree that a visual representation would aid interpretation. However, due to the limited availability of coronene-specific degradation data and the incomplete pathway annotation obtained through KAAS, it was not possible to construct a complete predictive pathway. To address this, we have included Supplementary Figure 5, which displays the partial xenobiotic degradation pathway derived from KEGG annotations. This figure outlines the identified components of the strain’s potential metabolic capabilities and visually indicates the gaps and hypothetical nature of the degradation route.

• We acknowledged the limitation of relying solely on computational annotation and discussed the need for transcriptomic, proteomic, and metabolomic studies, as well as gene knockout and heterologous expression experiments. These points are now explicitly included the Discussion Section (in the final paragraph).

12. The conclusion should emphasize the broader implications, such as potential applications in bioremediation and industrial use. Limitations and future research directions should be explicitly stated.

Our response:

We have revised the conclusion section of our manuscript. Specifically, we have now highlighted the potential practical applications of Halomonas elongata in bioremediation of marine and hypersaline environments contaminated with polycyclic aromatic hydrocarbons; Discussed the prospective industrial applications of secondary metabolites; indicated the current lack of experimental validation; Recommended specific future studies involving targeted genetic experiments (such as gene knockout and overexpression) and comprehensive metabolite profiling.

Reviewer #2

1. Introduction needs some more expansion specially for the recent work published worldwide.

Our response:

This comment is related to comment No 2 of the reviewer 1. We have expanded the Introduction section to incorporate more information on we have added relevant and appropriate references that the readers can you to gain insight on PAHs degradation. We have also added more information on the bacterial strain used in this study

2. Where bacteria collected from?

Our response:

As discussed earlier, this information has been added in the Introduciton.

3. Background of the bacteria lacking from the manuscript.

Our response:

This point, which as also raised by the Reviewer #1, has been addressed in the Introduction.

4. Detailed procedure needed about whole genome sequencing and quality assessment.

Our response:

This has been done (in Material & Methods).

5. Some headings/ subheading / paragraphs missing the major citation, please update accordingly.

Our response:

This has been checked and corrected.

6. There are several capitalizations between the sentence and heading. Authors should go throughout the manuscript for betterment of the article such as tile of fig 3, table 2 etc.

Our response:

We have followed the journal guidelines and the titles of figures and Tables have been crossed check and corrected.

7. Discussion part needs more expansion especially relation with the recent work published.

Our response:

This has been done. More discussion has been added.

8. References needs for formatting according the journal guidelines, there are several mistakes such as reference no 10 in the list. Fermentation 2022, Vol 8, 412 Page 260 2022;8:260

Our response:

This has been corrected now.

9. Clarity of the figures blurry

Our response:

The figures have been enhanced as soon as much we could, using available tools.

END DOCUMENTS

---

## [Decision Letter · Decision Letter 1]

30 Jul 2025

Dear Dr. Nzila,

Thank you for submitting your manuscript to PLOS ONE. After careful consideration, we feel that it has merit but does not fully meet PLOS ONE’s publication criteria as it currently stands. Therefore, we invite you to submit a revised version of the manuscript that addresses the points raised during the review process.

We look forward to receiving your revised manuscript.

Kind regards,

Bijay Kumar Behera, Ph.D.

Academic Editor

PLOS ONE

Journal Requirements:

Additional Editor Comments:

Dear Author,

Kindly address the reviewer comments and revised the manuscript accordingly.

Reviewers' comments:

Reviewer's Responses to Questions

**Comments to the Author**

Reviewer #1: (No Response)

Reviewer #2: All comments have been addressed

2. Is the manuscript technically sound, and do the data support the conclusions?

Reviewer #1: Partly

Reviewer #2: Yes

3. Has the statistical analysis been performed appropriately and rigorously?

Reviewer #1: N/A

Reviewer #2: N/A

4. Have the authors made all data underlying the findings in their manuscript fully available?

Reviewer #1: (No Response)

Reviewer #2: Yes

5. Is the manuscript presented in an intelligible fashion and written in standard English?

Reviewer #1: No

Reviewer #2: Yes

Reviewer #1: In Abstract "This study extends that work through whole-genome sequencing utilizing Oxford Nanopore Sequencing, a long-read sequencing technology, followed by bioinformatics analysis to uncover the genetic mechanisms enabling this bacterium to degrade coronene effectively." I still don't find the uncovered genetic/molecular mechanism for coronene degradation by the bacterium.

Though authors have used the data from their previous research, in my opinion, still there should be a section on the sample collection where the process can be explained in brief by citing to their previous paper.

Line 110 : "ONS technology" ONS should be expanded at its first instance.

"What is the basis of the cutoff 1e-06 in line 126?". My question was why it is 1e-06 not zero or any other value.

Authors failed to understand my comment on "Table 1 should be properly placed for easy understanding, statistics related to Q20/Q30 should be given as minimum Q20 is an widely accepted Phred Score. The N50 value should also be provided in this table." It's not about the location, it was about the rearrangement of the table in an easily readable way. Further, they have not presented the N50 value in the table. In ONS many manuscripts presented the % of reads above Q=20. I understand the value will not be as much as Illumina. However, presenting these scores gives a better idea on read quality.

I still think experimental validation of identified genes would strengthen claims about metabolic pathways and thereby providing insights into the molecular mechanism.

Reviewer #2: The authors have done excellent work. Congratulations to all the authors for the publication in the fantastic journal

**Do you want your identity to be public for this peer review?** For information about this choice, including consent withdrawal, please see our Privacy Policy

Reviewer #1: No

Reviewer #2: No

---

## [Author Response · Author response to Decision Letter 2]

10 Sep 2025

Rebuttal Letter

Manuscript ID: PONE-D-25-12285

Title: Whole Genome Sequencing, Characterization, and Analysis of a Coronene-Degrading Bacterial Strain, Halomonas elongate

We sincerely thank the reviewers for their time, effort, and constructive comments on our manuscript. We have taken note that Reviewer #2 found the revised version acceptable. Regarding Reviewer #1, some of his comments have already been discussed in our first revision, and in the rebuttal response below, we have carefully addressed all the concerns he has raised.

Reviewer #1

1. In Abstract: "This study extends that work through whole-genome sequencing utilizing Oxford Nanopore Sequencing, a long-read sequencing technology, followed by bioinformatics analysis to uncover the genetic mechanisms enabling this bacterium to degrade coronene effectively." I still don't find the uncovered genetic/molecular mechanism for coronene degradation by the bacterium.

Our response:

We thank the reviewer for this important observation. In the results section, we have clearly shown that our genomic analysis did not reveal a coronene degradation pathway. Instead, we identified genes associated with xenobiotic metabolism, suggesting that H. elongata may employ an alternative or novel degradation mechanism, and we have emphasized that this finding requires further experimental validation. To make this point clear, and in light of the reviewer's comment, we have summarised this information in this abstract.

2. Though authors have used the data from their previous research, in my opinion, still there should be a section on the sample collection where the process can be explained in brief by citing to their previous paper.

Our response:

“The same comment was raised during the first revision, and we addressed it by mentioning it in the Introduction. However, in light of the reviewer’s comment, we have now added this information in the Materials and Methods section under ‘DNA isolation, whole genome sequencing, and quality assessment.

3. "ONS technology" ONS should be expanded at its first instance.

Our response: This is done. It was explained in the Materials & Methods section, and has now been added to the abstract.

4. What is the basis of the cutoff 1e-06 in line 126?". My question was why it is 1e-06 not zero or any other value

Our response

This comment was raised and discussed in the previous rebuttal. However, in light of the review comment, we have further clarified it (Material and Methods section). Zero is not a valid cutoff for BLAST; thus, an e-value of 1e-06 was chosen as a stringent threshold that minimizes false positives while retaining biologically meaningful homologs.

5. Table 1 should be properly placed for easy understanding: statistics related to Q20/Q30 should be given as minimum Q20 is an widely accepted Phred Score. The N50 value should also be provided in this table." It's not about the location, it was about the rearrangement of the table in an easily readable way. Further, they have not presented the N50 value in the table. In ONS many manuscripts presented the % of reads above Q=20. I understand the value will not be as much as Illumina. However, presenting these scores gives a better idea on read quality.

Our response:

We thank the reviewer for this comment. Regarding Q20/Q30, as noted earlier, these metrics are Illumina-specific Phred quality scores and are not directly applicable to Oxford Nanopore sequencing data, which uses a different base-calling model and error distribution. Instead, ONT quality is evaluated through alternative measures (e.g., mean read quality, read length distributions, and base-calling accuracy), which we have already reported in Table 1. We have added q12 and q15 values.

Regarding the table readability, in preparing the previous revision, we organized Table 1 to present related parameters together (read quality, assembly statistics, and completeness) to allow straightforward interpretation. All critical information, including N50 and ONT-appropriate quality metrics, is provided in text, thus, we suggest to keep the current format.

As for the comment on NE50, the assembly N50 value has been included in Table 1.

6. I still think experimental validation of identified genes would strengthen claims about metabolic pathways and thereby providing insights into the molecular mechanism.

Our response:

The point raised by the reviewer is indeed relevant. However, as discussed in the manuscript, this study was designed as a computational genomic investigation using long-read sequencing and bioinformatics tools. We fully agree with the reviewer that experimental validation of the identified genes would strengthen our findings; however, such work was beyond the scope of the present study. As noted, further studies will be required to validate the proposed genes. We have clarified this point in the Discussion and Conclusion sections.

Reviewer #2

We thank Reviewer #2 for their supportive feedback and note that no further comments were raised.

---

## [Decision Letter · Decision Letter 2]

28 Sep 2025

Whole Genome sequencing, Characterization and Analysis of coronene degrading Bacterial strain Halomonas elongata

PONE-D-25-12285R2

Dear Dr. Nzila,

We’re pleased to inform you that your manuscript has been judged scientifically suitable for publication and will be formally accepted for publication once it meets all outstanding technical requirements.

Kind regards,

Bijay Kumar Behera, Ph.D.

Academic Editor

PLOS ONE

Additional Editor Comments (optional):

Dear Dr. Nzila,

I am pleased to inform you that your manuscript entitled "Whole Genome sequencing, Characterization and Analysis of coronene degrading Bacterial strain Halomonas elongata" (Manuscript Number: PONE-D-25-12285R2) has been accepted for publication in PLOS ONE.

Following rigorous peer review and careful editorial assessment, your article has been found to meet the journal’s publication criteria for scientific rigor, originality, and relevance. The study provides valuable new insights into the genetic and functional aspects of coronene-degrading Halomonas elongata, and we are confident that it will be of broad interest to the scientific community.

Your manuscript has now moved to the production process. You will soon receive further instructions regarding copyediting, proofing, and publication details from the PLOS ONE team.

Congratulations on your achievement, and thank you for choosing PLOS ONE as the venue for your research.

Sincerely,

Dr. Bijay Kumar Behera

Academic Editor

PLOS ONE

Reviewers' comments:

Reviewer's Responses to Questions

**Comments to the Author**

Reviewer #1: All comments have been addressed

2. Is the manuscript technically sound, and do the data support the conclusions?

Reviewer #1: (No Response)

3. Has the statistical analysis been performed appropriately and rigorously?

Reviewer #1: Yes

4. Have the authors made all data underlying the findings in their manuscript fully available?

Reviewer #1: Yes

5. Is the manuscript presented in an intelligible fashion and written in standard English?

Reviewer #1: Yes

Reviewer #1: Though all my comments has not been satisfactorily addressed, authors addressed majority of my concerns. Understanding the authors limitaions for validation, I recommend the manuscript.

**Do you want your identity to be public for this peer review?** For information about this choice, including consent withdrawal, please see our Privacy Policy

Reviewer #1: No

---

## [Editor Report · Acceptance letter]

PONE-D-25-12285R2

PLOS ONE

Dear Dr. Nzila,

I'm pleased to inform you that your manuscript has been deemed suitable for publication in PLOS ONE. Congratulations! Your manuscript is now being handed over to our production team.

Kind regards,

on behalf of

Dr. Bijay Kumar Behera

Academic Editor

PLOS ONE